# How do users of a 'digital-only' contraceptive service provide biometric measurements and what does this teach us about safe and effective online care? A qualitative interview study

Paula Baraitser [1,2] Hannah McCulloch,[2] Alessandra Morelli [2] Caroline Free[3]

¹SH:24, London, UK
²King's Centre for Global Health and Health Partnerships, King's College London, London, UK
³Public Health interventions Unit, London School of Hygiene and Tropical Medicine, London, UK

**Correspondence to**
Dr Paula Baraitser;
paula_baraitser@mac.com

## ABSTRACT

**Objectives** To describe user experience of obtaining and uploading biometric measurements to a 'digital-only' contraceptive service prior to a prescription for the combined oral contraceptive (COC). To analyse this experience to inform the design of safe and acceptable 'digital-only' online contraceptive services.

**Setting** An online contraceptive service available free of charge to women in South East London, UK.

**Participants** Twenty participants who had ordered the combined oral contraceptive (COC) online. Our purposive sampling strategy ensured that we included participants from a wide range of ages and those who were and were not prescribed the COC.

**Intervention** A 'digital-only' contraceptive service that prescribes the COCafter an online medical history and self-reported height, weight and blood pressure (BP) with pills prescribed by a GMC registered doctor, dispensed by an online pharmacy and posted to the user.

**Design** Semistructured interviews with a purposive sample of 20 participants who were already enrolled in a larger study of this service.

**Analysis** Inductive, thematic analysis of the interviews assisted by NVivo qualitative analysis software.

**Results** Users valued the convenience of 'digital-only care' but experienced measuring BP but not height or weight as a significant barrier to service use. They actively engaged in work to understand and measure BP through a combination of recent/past measurements, borrowed machines, health service visits and online research. They negotiated tensions around maintaining a trusting relationship with the service, meeting its demands for accurate information while also obtaining the contraception that they needed.

**Conclusion** Digital strategies to build trusting clinical relationships despite a lack of face-to-face contact are needed in 'digital-only' health services. This includes acknowledgement of work required, evidence of credible human support and a digital interface that communicates the health benefits of collaborating with an engaged clinical team.

## BACKGROUND

People have always acted to maintain and improve their own health, but self-care within formal health systems is a recent development.[1 2] Early research and interventions on self-care within clinical systems focused on shared decision-making and symptom management for those living with long-term conditions.[2 3] More recently, complex self-care tasks, for example, blood sugar monitoring[4] or self-testing for sexually transmitted infections,[5] are included. These tasks often happen outside clinical contexts with remote clinical support and require access to clinical technologies (such as blood sugar monitors or lancets to collect blood) and digital interfaces that support technology use by storing data, providing information and facilitating remote communication between users and clinicians.

National and international health policy documents suggest that digitally enabled, self-care offers the potential to improve user experience with more convenient, more informative and more cost-effective services.[6–9] There is a growing body of literature on digital health services that support home self-monitoring of those with long-term conditions such as diabetes or hypertension as an adjunct to healthcare from a face-to-face

### Strengths and limitations of this study

► This is the first study that looks at user experience of online contraceptive access.
► Through qualitative interviews, it provides detailed accounts of the work required to negotiate a relationship with a 'digital-only' service.
► The analysis identifies elements of user relationships with digital services that support accurate reporting of biometric measurements.
► Like all qualitative research, it offers in-depth accounts from small numbers of people and may not be representative of larger populations.

service[10 11] and on digital self-monitoring of biometric data (self-tracking) that occur outside the health system. This literature suggests that outcomes of transferring clinical technologies to domestic spaces may be unpredictable as they are adapted by users for home settings, and framed within non-clinical understandings of health and healthcare.[10–15] There is less research on digitally facilitated self-care as a stand-alone intervention without a 'face-to-face' element.[16] This gap in the evidence has implications for the 'digital-only' or 'digital-first' services that are increasingly part of health policy.[9]

Sexual health services have been early adopters of digitally enabled self-care,[17] possibly because self-care is particularly attractive where stigma is a barrier to service access.[1] Online testing for sexually transmitted infections is now routine practice in many public health systems[5 18 19] and there is an expanding market for online contraception.[20] These services are 'online-only' services where it is anticipated that the whole clinical pathway will be delivered online.

We studied a UK-based, online sexual health service (SH:24) providing the combined oral contraceptive (COC), delivered home, by post after completion of a medical history and self-reporting of height, weight and blood pressure (BP). As an early example of 'digital-only' facilitated self-care, we were interested in user experience of measuring and uploading BP, height and weight readings prior to online prescriptions. This is an example of 'pushed' self-tracking[10] or 'clinical' self-tracking,[11] where the service requires self-measurement as a condition of service delivery.

The measurement of BP and body mass index (BMI) prior to first prescription of COC is recommended by national and international guidance.[21 22] Ethinyloestradiol (the oestrogenic component of all COCs) stimulates an increase in hepatic protein production including angiotensinogen leading to a slightly reduced renal blood flow, and therefore a slight increase in BP.[23] In about 2% of women, BP may increase significantly after starting the COC.[24] Since hypertension increases risk of cardiovascular disease, BP is monitored before first prescription of the pill and then annually, and the COC is contraindicated if BP is consistently above 140/90 mm Hg. Obesity is similarly associated with risk of cardiovascular disease, and height and weight are also measured prior to COC prescription. If BMI (weight in kilograms divided by height squared in metres) is equal to or greater than 35 kg/m$^2$ then the risks of taking the pill are considered to outweigh the benefits.[21] In face-to-face clinical practice, biometric measurements are collected and recorded by healthcare professionals prior to prescribing, usually as a single measurement with repeat readings taken if the first reading is abnormal. In the online service, users are asked to upload a recent reading.

Accuracy of biometric readings in this context is clinically important and may be influenced by user experience of this process.[10] We were, therefore, interested in how users experience the process of self-reported measurement and how this influences the validity of the readings they upload. In addition, we were interested in how experience of self-reported measurement and accurate reporting might be facilitated by a digital interface. Our research question was 'How do users of a 'digital-only' contraceptive service provide biometric measurements and what does this teach us about safe and effective online care?

## INTERVENTION
We studied user experience within a free, online service providing the COC within a diverse area of South East London, UK, with high rates of socioeconomic deprivation and poor sexual and reproductive health indicators, such as high rates of emergency contraception () use and under-18 conceptions.[25]

The service was advertised through social media, print media (cards and stickers) and local clinics. The digital interface of the online service invites users to order the COC after completing a medical history mapped to national guidelines[21] and report their height, weight and BP. Users are informed that a BP reading is required at the start of the order journey and basic information on the risk of taking the COC with high BP is provided (online supplemental appendix A). The service provides advice on how to obtain a BP measurement from the machines available in general practice waiting rooms and pharmacies or by using a home BP machine. Those who have ordered the COC from the service before are reminded 2 weeks prior to the end of their pill supply to obtain a BP reading in advance of their next order (text message sent to users to remind them to check their blood pressure before they order their next supply of pills. You have 2 weeks' supply left of your oral contraception. You can repeat order here: https://sh24.org.uk/coc_orders/new. To successfully repeat you will need to know bloodpressure. If you do not know your blood pressure you can find out how to check it here: https://sh24.org.uk/contraception/order#question_7. Text back if you would like help. Thanks, SH:24). Users who upload a BP reading outside the normal range are contacted by the prescribing clinician for further details on where and how it was measured and to request additional measurements if appropriate. Users and clinicians communicate mainly by text message and the tone of the clinical communication is neutral and polite, for example, 'Hello, thank you for your order, your contraception will be dispatched today, to this address by Royal Mail tracked but not signed for, many thanks, SH:24'. Where the prescribing clinician finds that a BP or BMI that is higher than the levels required for safe prescribing that are specified in national guidance then users are denied the COC and signposted to information about the alternatives including the option to order the progestogen only pill which does not have the same contraindications. This pathway requires 'work' done by the three entities involved in this process, the service user,

## User

Understands requirement for BP, height and weight.

Accesses information on BP height and weight as required.

Accesses a BP and BMI measurement.

Assesses whether the measurement is normal.

Uploads the readings when requested within the electronic order journey.

Communicates with the clinician as necessary.

## Digital interface

Requests information on BP, height and weight.

Presents information on BP and BMI including normal values and the risks of taking the COC with high BP or BMI.

Stores the BP or BMI on the clinical record and highlights it when outside the normal range to draw the attention of the clinician

Enables communication between service user and provider through text message or telephone.

Records the communication for reference.

## Clinician

Completes a clinical assessment of any readings that are highlighted within the system.

Communicates with the service user if the readings are outside the normal range.

Assesses the validity of the readings based on the communication with the user.

Requests additional measurements if appropriate.

Decides whether to prescribe the COC or not.

Effectively communicates this decision

**Figure 1** The work done by the user, digital interface and clinician to support communication of accurate biometric measurements. BMI, body mass index; BP, blood pressure; COC, combined contraceptive pill.

the digital interface and the prescribing clinician. This work is described in figure 1.

Contraindications to the COC are classified within national and international guidelines on a scale of 1–4. A contraindication classified as 1 or 2 suggests that the benefits of use outweigh the risks and a contraindication classified as 3 or 4 suggests that the risks outweigh the benefits.[21] The COC is not prescribed within this service to women reporting category 3 or 4 contraindications. Prescriptions are authorised by a UK General Medical Council registered doctor are sent electronically to a UK registered online pharmacy, which dispenses the medication and posts it to the service user. The service is registered with the Care Quality Commission and is compliant with National Health Service digital standards.

## METHODS

The nested qualitative study reported here aimed to understand user experience of obtaining and uploading biometric measurements prior to online contraceptive prescriptions and was part of a larger study that compared self-reported height, weight and BP measurements submitted to an online contraceptive service with researcher-measured values.

Semistructured interviews with participants were completed by HM, a white British, 26-year-old female researcher, who has used COC. The interview guide started with general questions on first and subsequent experience of accessing contraception, followed by a detailed account of use of the online service with a particular focus on experience of measuring and reporting height, weight and BP. Interviews were completed in a private room in a university building and lasted between 23 and 83 min with a median length of 40 min and were audio recorded and transcribed verbatim.

## Public and patient involvement (PPI)

Patients and the public were involved in identifying the importance of this research question. We used existing, nationally significant PPI conducted by the James Lind Alliance and endorsed by the UK Faculty of Sexual and Reproductive Health Care, which found that among the general population 'research to evaluate which interventions increase uptake and continuation rates of effective contraceptive methods' was the number one priority for academic research on contraception in the UK (https://www.fsrh.org/about-us/fsrh-committees/contraception-priority-setting-partnership-psp-in-conjunction/). In addition, we completed our own research priority setting exercise involving patients and the public prior to developing the research proposal based on two focus groups with service users that identified research into online service delivery as a priority topic for research on contraception. The logic model for the intervention was similarly developed with input from patients and the public including interviews with 21 stakeholders. Intervention development involved over 100 service users through a process of human-centred design.

## ANALYSIS

We completed an inductive, thematic analysis assisted by NVivo qualitative analysis software (V.12). We read and reread the interviews (PB, HM and AM) to develop themes and completed an iterative process of four rounds of theme refinement, rechecking our data for material that refuted, supported or developed our themes and recoding with discussion among the whole research team to reach consensus at each stage. Our final coding strategy is provided in online supplemental appendix B.

**Table 1** Sociodemographic characteristics and self-reported contraceptive history of participants

| | Study participants of quantitative study from which the qualitative sample was identified, n=365 (%) | Study participants of nested qualitative sample, n=20 |
|---|---|---|
| Age (years) mean | 24.76 | 24.89 |
| 18–19 | 29 (7.95) | 2 |
| 20–24 | 175 (47.95) | 13 |
| 25–34 | 147 (40.27) | 2 |
| 35+ | 14 (8.84) | 3 |
| Ethnicity | | |
| White English/Welsh/Scottish/ Northern Irish/British/other | 221 (60.55) | 12 |
| Black African/Caribbean/ British/other | 57 (15.62) | 5 |
| Asian/Asian British | 33 (9.04) | 0 |
| Mixed/multiple ethnic groups | 42 (11.51) | 2 |
| Other ethnic groups | 10 (2.74) | 1 |
| Not known/prefer not to say | 2 (0.55) | 0 |
| Index of Multiple Deprivation of postcode of residence | | |
| 1 (most deprived) | 131 (36.19) | 4 |
| 2 | 148 (40.88) | 9 |
| 3 | 66 (18.23) | 5 |
| 4 | 13 (3.59) | 2 |
| 5 (least deprived) | 4 (1.10) | 0 |
| Qualifications | | |
| No academic qualifications | 1 (0.27) | 0 |
| GCSES (General Certificate of Secondary Education, or equivalent level) | 13 (3.56) | 1 |
| AS/A levels (Advanced level or equivalent) | 34 (9.32) | 2 |
| Higher education qualifications (or equivalent level) | 316 (86.58) | 17 |
| Not sure or other | 1 (0.27) | 0 |

## RESULTS

Table 1 shows the sociodemographic characteristics of the study population and the nested qualitative study population. Our respondents were aged between 19 and 37 years and most had participated in higher education.

The themes that emerged during analysis not only related to the convenience of 'digital-only care' but also to the measurement of BP, but not height or weight, as a barrier to service use. Users acknowledged the work required to fulfil the requirements of the service and actively engaged with the need to understand and measure BP. They negotiated tensions around maintaining a trusting relationship with the service, meeting its demands for accurate information while also obtaining the contraception that they needed at the time that they needed it.

Our participants reported difficulty accessing the CCP from face-to-face services were concerned that a gap in their pill supply would put them at risk of pregnancy and were relieved to find that a free online service existed. This added a sense of urgency to their need to access contraception.

> Say it was like Friday, I needed the pill by the Monday so I was like panicking, what shall I do? Where do I go now that I tried everywhere and nowhere seemed available? So, yeah, I just looked online like and then just found that you could order it online…… I felt relieved that I could actually get it Participant F

In a context where they felt access to contraception was difficult, they employed a range of strategies within both online and face-to-face care to mitigate this.

> Basically, I needed to get a top-up for more pills… Because I finish work at 6:00 and I start at 9:00, is the perfect hours for not being able to go to a clinic. My sister finishes work at four o'clock. I asked my sister, "Can you get me a top-up please and I'll pay you because I really need it." Like, I'll pay her for her time having to wait in the clinic. Participant C

Participants felt that the CCP was a low-risk medication that was appropriately accessed online. However, as they engaged with the ordering process, the work to measure BP, but not the work to measure height and weight, or report medical history, was seen as a significant barrier to access.

> With your blood pressure, there's just no way to tell so it's just very different to someone saying, 'What's your height? What's your weight?' because you can sort of look at yourself in the mirror and guess, but you just can't do that with your blood pressure. Participant D

Although most users uploaded a BP measurement, they lacked information on normal values, variation over time or the relationship of BP to feeling healthy or stressed. It was clear that repeated use of face-to-face services to obtain the CCP, including regular BP measurements, often over many years, had not provided sufficient knowledge about BP or its relation to CCP use to understand a specific BP measurement although it did provide users with a sense of whether their BP was within the normal range.

> like when you go to the clinic to take your blood pressure, no-one tells you what it is. They just take it and then they write it down. No-one says, 'Oh, for your information, your blood pressure's this'. Even if you're like high, you would normally have to say, 'Oh, what was my high?' Participant D

Although the recommendation from the online service was to obtain a BP measurement at a pharmacy, most participants did not recall this advice or follow it except one who was friend with her local pharmacist and obtained her BP reading from this source. The need to visit a clinical setting for a BP check was seen as significantly reducing the convenience of the service.

> For me, that kind of takes away the purpose of the service, because I want to use the service to avoid having to go to the pharmacy, because I just want to be quick and do it… Participant M,

Participants felt that the work to obtain a BP measurement was insufficiently acknowledged by the digital interface, while the work to access height and weight measurements was acceptable and required no further support. Social networks were an important source of access to BP machines, for example, use of a machine owned by an older relative or friend or using existing healthcare professional contacts within social networks. Research was required to understand BP and its measurement. This was mainly online, for example, searching for normal BP values and researching the health consequences of high BP. Although service users were advised to obtain a BP reading from their local pharmacy, most did not feel that this was a useful option for them. Many participants did not have access to a BP monitor and combined strategies to generate what they felt was an accurate measurement, increasing their confidence in the value they uploaded by triangulating between different methods of measurement. Their accounts of this (see table 2) illustrate the disjuncture between the expectations from the online service, which required the systolic and diastolic readings, and the experience of users, who often knew that their BP was normal but lacked access to the specific readings.

As a result of the difficulty accessing BP measurements many participants found themselves negotiating their relationship with this new type of clinical service as they sought to understand whether this was a relationship of trust that required honest reporting of biometric measurements. They reflected on the whether their information would be reviewed by a clinician or a digital algorithm. They felt that communication with a clinician would require greater commitment to providing an accurate reading and they worked to understand what might be a reasonable mix of measurement techniques to provide such a reading if they did not have access to the recent measurement requested by the service. They saw the need to provide BP as something that was required by the service, and therefore a barrier to accessing contraception but also as something that might be important for their own health. Some users took on the new responsibility very seriously and took personal responsibility for providing an accurate measurement.

> I just suddenly had like a fear of god moment, where I was like, if I lie about this and then something does happen maybe they'll turn around and be like "well, you're not supposed to be over this weight or you're blood pressure has got to be this amount"…… things with your health, you've got to take seriously. It makes me nervous. So, even if it is the pill, which 9/10 isn't going to be an issue… Because I wasn't- there wasn't a face to it, I felt I really needed to make sure I was doing it right Participant L

Others felt that this responsibility was shared with the online service and expected the clinicians/algorithms to identify inaccurate readings and uncover inappropriate strategies to collect them. For example, one participant who had guessed a reading that was biologically impossible

| Table 2 | Work required to generate a BP measurement though a combination of strategies | |
|---|---|---|
| Case details | Measurement strategy | Account of the measurement process |
| Participant F | Remembered an old reading, knew that BP was usually normal and planned to get it checked before the next order | I think that was like from a doctor's that I had, like I think a few months beforehand. So, it was within normal range so it was fine and just remembered it …. so I knew that it wouldn't be like as reliable as doing a blood pressure straight away but within that, like I needed it quickly so I knew that normally like I'm fine with blood pressure so I just put in what I could remember from the last one, like it was normal …. So I think, if I was to like order again, I think I would maybe consider actually taking it like closer to the point |
| Participant O | Remembered that a previous reading had been normal and looked up a normal value online | Yeah, when the blood pressure thing came up I was a bit nervous … I knew that I had healthy blood pressure, so I could kind of estimate… for me I just sort of, googled what a healthy kind of reading was and vaguely remembered from my last time, anyway |
| Participant T | Estimated BP based on multiple previous readings from GP (general practitioner) and home BP machine | So, the blood pressure I used an estimate of what the previous readings I'd had in the last twelve months. I'd had a GP visit and I had it checked. And then I'd had it checked a few times at home, 'cause we had a blood pressure monitor at home, that we were trying out and seeing if it worked or not. And all of those had been fine in the past. So I was fairly confident that it wouldn't have changed |

BP, blood pressure.

and was contacted by a clinician from the service to find out when and how it was measured. She was then surprised and a little angry when her (untrue) explanation was accepted, feeling that the clinician should have been more questioning of the information she gave.

> The first message that came up was like "Oh you will be contacted by a nurse or something. To discuss… further" … I got a text message from the nurse saying how did you gather your blood pressure information. And I was like "Oh when I went to the clinic, she wrote it down for me". But I just entered it wrong…. And they said "Oh OK then, that's fine. Thank you. We'll let the nurse know…. I thought it was going to be much more nerve-wrecking, like calling my phone … Participant G

This participant was prepared to compromise her health in order to obtain the contraception that she needed.

> Then I just had to google it, I just had to google "What is the correct blood pressure". And then one came up, they gave me a average. And I was like OK. Put that in… Participant G

Participants wanted more information about BP and its importance in relation to the CCP and more help with obtaining a reading although none of the participants reported requesting help from the service. The digital interface seemed to communicate the BP measurement as the users' responsibility and the inputting of normal measurements as requirement. Participants felt that making contact might compromise their entitlement to the CCP and none did so unless the service contacted them first.

Throughout the interviews there was a marked difference between participants' description of measuring BP and their experience of measuring height and weight. They felt that monitoring their weight was a common health maintenance activity and they had been previously encouraged to do this. Height does not change in adulthood and most people have had it measured at some time and remember this measurement at least approximately.

## DISCUSSION

The need to measure BP was experienced as an important barrier to access an otherwise convenient online contraceptive service. Significant work was required by users, the digital interface and clinicians to facilitate the process of BP measurement, communication and assessment. Users completed social, ethical and information gathering work to provide numerical BP values often combining strategies to maximise the validity of the reading uploaded, researching what is normal and negotiating what they felt was an appropriate commitment to providing accurate information. This negotiation referenced their need to access the pill urgently, feelings about a trusting relationship with the clinical service and concerns about the health implications of taking the pill with high BP. Better communication of the presence of the clinicians within the service and their willingness to support users would have made this process easier. Users found the measurement of height and weight straightforward and did not experience this as a barrier to access the service in any way.

Developing capacity to use effective, evidence-based self-care for sexual and reproductive health is a public good[1] and may expand access to contraception,[20] abortion[26] and testing for sexually transmitted infections.[5] However, self-care within health systems involves delegation and redistribution of health work so that while it may allow people greater control over their lives it also brings new responsibilities[27] and new requirements for clinical support.

Whereas health policy discourses have tended to portray the digital interfaces that mediate digital healthcare as neutral channels with predictable outcomes,[28] this study and others[15 29] suggest that digital interfaces do important work in brokering (or not) relationships of trust within online clinical services and that these will influence user experience and clinical outcomes.[28 30] New media for consultations inevitably change some aspects of clinical relationships[31] and at present very little is known about the 'rules of engagement' for the management of clinical consultations through online portals.[16 32] Our study suggests that it is important that both user and provider are clearly present as individuals within the online communication process, that provider commitment to building a clinical relationship is clearly demonstrated and visible to the service user and that the health benefits of accurate reporting of biometric measurements are clear.

Self-monitoring BP has become popular among those with long-term conditions and GPs estimate that almost a third of patients with hypertension self-monitor their BP.[33] Similarly, research among pregnant women with gestational hypertension shows that self-measurement with support and training is acceptable and empowering.[34 35] However, experience of self-measurement of BP for the population of young contraceptive users in our study seems to be quite different as their need to self-measure is intermittent, they lack information and received no training on BP and its measurement.

## CONCLUSIONS

Our findings support the view that digitally facilitated self-management will require close collaboration between users and providers, particularly in 'digital-first' services. The digital interface is a particularly important in communicating and supporting this collaboration and its importance is often underestimated. As digitally facilitated self-management becomes increasingly prevalent within health systems, we suggest that the digital interfaces that support them should:

1. Fully acknowledge the work that users, providers and digital interfaces do within online services.

2. Ensure that user and provider are clearly present as individuals within the online communication process, that provider commitment to building a clinical relationship is demonstrated and that the health benefits of the provision of accurate information are communicated.

3. Enable appropriate adaptation of clinical processes outside clinical contexts to facilitate self-care, for example asking whether BP was normal rather than asking for specific values.

The service evaluated is currently being redesigned in response to this research aiming to communicate better information about BP and its measurement, realistic strategies for BP measurement and a commitment to collaborative user/provider relationship.

**Contributors** PB contributed to the conception and design of the study, oversaw data collection, contributed to data analysis and drafted the manuscript. HM contributed to data collection and analysis and contributed to the manuscript on the manuscript. AM contributed to data collection and analysis and commented on the manuscript. CF contributed to the conception and design of the study, contributed to data analysis and commented on the manuscript.

**Funding** This report is independent research funded by the National Institute for Health Research (NIHR) (Research for Patient Benefit Programme, PB-PG-0815-20009). The views expressed in this publication are those of the author(s) and not necessarily those of the NIHR or the Department for Health and Social Care.

**Competing interests** PB is clinical director of the online service studied and is one of the prescribers within this service. Since completion of this study, HM has worked at the online service on a different research project. This work is part of a process of research on innovation at SH:24 where all innovation is the subject of research to ensure that learning is shared widely. SH:24 is a 'not-for-profit' organisation that provides health services to the National Health Service and sharing learning through innovation and research is one of the principles of the organisation.

**Patient and public involvement** Patients and/or the public were involved in the design, or conduct, or reporting, or dissemination plans of this research. Refer to the Methods section for further details.

**Patient consent for publication** Not required.

**Ethics approval** Ethical approval for the study was granted by the East Midlands—Leicester Research Ethics Committee (ref 17/EM/0181) and all study participants provided informed consent.

**Provenance and peer review** Not commissioned; externally peer reviewed.

**Data availability statement** Data are available upon reasonable request. The anonymised qualitative interviews are available from the authors on request.

**ORCID iDs**
Paula Baraitser http://orcid.org/0000-0002-3354-6494
Alessandra Morelli http://orcid.org/0000-0002-9803-2136

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
