## [Reviewer comments · BMJ Open]

ARTICLE DETAILS

TITLE (PROVISIONAL)	How do users of a 'digital-only' contraceptive service provide biometric measurements and what does this teach us about safe and effective online care? A qualitative interview study
AUTHORS	Baraitser, Paula; McCulloch, Hannah; Morelli, Alessandra; Free, Caroline

VERSION 1 – REVIEW

REVIEWER	Carrie Purcell MRC/CSO Social and Public Health Sciences Unit, University of Glasgow
REVIEW RETURNED	27-May-2020

GENERAL COMMENTS	This is an interesting and timely contribution to understandings of telemedicine and self-management of reproductive (and other forms of) healthcare. The service evaluation presented offers potentially useful evidence on barriers to contraceptive access. The comments below are minor, with a view to enhancing the overall clarity of the paper. Methods: I'm not clear why there is a description of the intervention development methods here (e.g. reference to a logic model). It may be interesting to the reader to include this information, but it should be clearly separated out from the methods for the evaluation study on which the paper focuses. I would clarify Table 1 to present the qualitative study sample, and then also present the corresponding data on the wider study. Or, if you prefer to present the findings as being from a 'nested qualitative study', then frame it as such from the outset. Results This section needs some initial set up to outline what the identified themes were, before you begin to present the analysis. It is not noted until p8 exactly how service users were expected to obtain a BP reading – worth explaining up front. It would also be useful to the reader if you could clarify (ideally early on) why some women would be / were not provided with a prescription – were they refused on grounds of the data provided? Discussion It strikes me as slightly odd that you do not introduce Fig 1 until the Discussion. If you aim for it to offer an illustration of the intervention, it might sit better in an early section. Linking it here to a summary of findings doesn't quite work, since it offers findings (on the work done by HPs) on which you do not present data.
--

	Lastly, the manuscript needs a thorough proof read for punctuation.
--	---

VERSION 1 – AUTHOR RESPONSE

Responses to reviewers

Many thanks for your helpful comments on this manuscript. We have revised it in line with your comments as follows:

Comments	Response
Please revise the 'Strengths and limitations' section of your manuscript (after the abstract). This section should contain five short bullet points, no longer than one sentence each, that relate specifically to the methods. The results of the study should not be summarised here.	We have revised the 'strengths and limitations' section to include short bullet points and removed any reference to results of the study.
I'm not clear why there is a description of the intervention development methods here (e.g. reference to a logic model). It may be interesting to the reader to include this information, but it should be clearly separated out from the methods for the evaluation study on which the paper focuses.	We have clarified that the references to intervention development and the importance of the research topic relate to the patient and public involvement that was part of the development of the intervention and the decision to study it. We have now marked this as a sub- heading entitled 'Patient and Public Involvement'
I would clarify Table 1 to present the qualitative study sample, and then also present the corresponding data on the wider study. Or, if you prefer to present the findings as being from a 'nested qualitative study', then frame it as such from the outset.	We have clarified table 1 and re-presented this study as a nested qualitative study within a larger observational study elsewhere in the manuscript
This section needs some initial set up to outline what the identified themes were, before you begin to present the analysis.	We have added a paragraph at the start of the results section to outline the identified themes before we present the analysis.
It is not noted until p8 exactly how service users were expected to obtain a BP reading – worth explaining up front. It would also be useful to the reader if you could	We have added a section to the description of the intervention to explain exactly how service users were expected to obtain a BP reading and

clarify (ideally early on) why some women would be / were not provided with a prescription – were they refused on grounds of the data provided?	to clarify why some women would be/were not provided with a prescription.
It strikes me as slightly odd that you do not introduce Fig 1 until the Discussion. If you aim for it to offer an illustration of the intervention, it might sit better in an early section. Linking it here to a summary of findings doesn't quite work, since it offers findings (on the work done by HPs) on which you do not present data.	We have moved figure 1 to the section on the intervention.
Lastly, the manuscript needs a thorough proof read for punctuation.	We have proof read the manuscript for punctuation.

VERSION 2 – REVIEW

REVIEWER	Carrie Purcell MRC/CSO Social and Public Health Sciences Unit, University of Glasgow, UK
REVIEW RETURNED	06-Jul-2020
GENERAL COMMENTS	The issues raised in the initial review have been adequately addressed, thank you.